# DETERMINANT REGULARIZATION FOR DEEP METRIC LEARNING

## ABSTRACT

Distance Metric Learning (DML) aims to learn the distance metric that better reflects the semantically similarities in the data. Current *pair-based* and *proxy-based* methods on DML focus on reducing the distance between similar samples, while expanding the distance of dissimilar ones. However, we reveal that shrinking the distance between similar samples may distort the feature space, increasing the distance between points of the same class region, and therefore, harming the generalization of the model. The regularization terms (such as $L_2$-norm on weights) cannot be adopted to solve this issue as they are based on linear projection. To alleviate this issue, we adopt the structure of normalizing flow as the deep metric layer and calculate the determinant of the Jacobian matrix as a regularization term that helps in reducing the Lipschitz constant. At last, we conduct experiments on several *pair-based* and *proxy-based* algorithms that demonstrate the benefits of our method.

## 1 INTRODUCTION

Deep metric learning (DML) is a branch of learning algorithms that parameterizes a deep neural network to capture highly non-linear similarities between images according to a given semantical relationship. Because the learned similarity functions can measure the similarity between samples that do not appear in the training data set, the learning paradigm of DML is widely used in many applications such as image classification & clustering, face re-identification, or general supervised and unsupervised contrastive representation learning Chuang et al. (2020). Commonly, DML aims to optimize a deep neural networks to span the projection space on a surface of hyper-sphere in which the semantic similar samples have small distances and the semantic dissimilar samples have large distance. This goal can be formulated as the discriminant criterion (and its many variants that appear in the literature) we summarize as follows.

$$\max\{d_\theta(\mathbf{x}_i, \mathbf{x}_j) | j \in \mathcal{S}_i\} < \delta_1 < \delta_2 < \min\{d_\theta(\mathbf{x}_i, \mathbf{x}_l) | l \in \mathcal{D}_i\} \qquad (1)$$

where $\theta$ are the parameters of the deep metric model, $\delta_1$ and $\delta_2$ are two tunable hyperparameters, and $\mathcal{S}_i$ and $\mathcal{D}_i$ are the sets of similar and dissimilar samples of the query $\mathbf{x}_i$, respectively.

Commonly the log-exp function $q_\lambda(\theta) = log(\sum_{j=1}^n e^{\lambda a_i(\theta)})$ Oh Song et al. (2016) is used to define the objective function in DML. Besides the definition of the objective function, many works point out that the performance of DML crucially depends on the informative sample mining (HSM) procedure and therefore focus their research direction on improving the HSM. Unfortunately, the explicit definitions of informative samples is still unclear, and the problem seems to be unsolved. This leads us to the following question: **what is the real reason that makes DML model so crucially depend on hard sample mining?** In this paper, we try to answer this question studying the local Lipschitz constant of the learned projection $f_\theta(\mathbf{x})$.

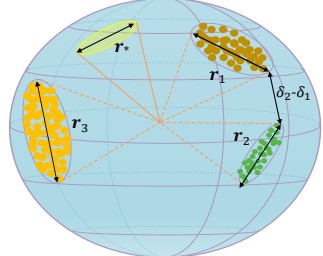

Figure 1: The illustration represents the feature space spanned by $f_\theta(\mathbf{x})$ learned by deep metric learning. $r_i$ is the radius $i$-th class of samples in training dataset and $r_*$ the radius of an unknown class. $\delta_2 - \delta_1$ reflects the distance between the two closest samples in class 1 and class 2.

Many recent advances have been presented on DML in many new excellent works Wang et al. (2019); Kim et al. (2020); Roth et al. (2022); Wang & Liu (2021); Schroff et al. (2015); Sun et al. (2020); Deng et al. (2019); Wang et al. (2018) since the first DML model was proposed in 2015 Schroff et al. (2015). Current methods present good generalization i.e., the learned metric can work well on the unseen classes. We attribute the good generalization performance of DML models to the fact that the learned function $f_\theta(\mathbf{x})$ extends in a continuous manner on the projecting region presenting a small Lipschitz constant w.r.t. the origin sample space. As we know, DML reduces the distances between similar samples which also reduces the local Lipschitz constant of $f_\theta(\mathbf{x})$ of the region surrounded by training samples. The continuity of $f_\theta(\mathbf{x})$ prevents the Lipschitz constant from changing fast, so $f_\theta(\mathbf{x})$ presents a small Lipschitz constant at points close the region where training samples are located. The learned projection having small Lipschitz constant induces a smaller upper bound on the empirical loss, which results on a better generalization performance. This interpretation explains why the projection $f_\theta(\mathbf{x})$ learned by distance metric learning has a good generalization even on unseen classes.

However, DML presents some drawbacks. Shrinking the cluster of each class increases the local Lipschitz constant of $f_\theta(\mathbf{x})$. This phenomenon can be divided into two effects. The first one corresponds to the increase of the Lipschitz constant caused by enlarging the distances between dissimilar samples, which was found by Song et al. (2021). This effect increases the local Lipschitz constant of $f_\theta(\mathbf{x})$ at the region between classes. The author of Song et al. (2021) commented that the failure of training triplet loss without semi-hard sample mining can be attributed to it. In the second case, the regions with large Lipschitz constant are a priori unknown. This effect may occur in the region that belongs to unknown regions or regions that belong to a cluster for a class. In the first case, the negative effect can be alleviated reducing the distance between dissimilar samples. Based on this strategy, Song et al. (2021) designs an KKN decision boundary to formulate a general framework of distance metric learning and proved that current state-of-the-art algorithms such as lifted structure loss, multi-similarity loss, circle loss or N-pair loss are special cases of it. Those implicitly mean that the sample mining strategies are designed to reduce the Lipschitz constant of the learned projection.

Regarding the second case, there are a few works that address this issue. A common strategy is to assign the position of centers of each class by minimizing designed energy functions Duan et al. (2019); Liu et al. (2018). The main assumption of this method is that if distances between centers are large, there is no need to shrink each class too much while still preserving the classes gap. However, this method has only positive results if it increases the Lipschitz constant at the region of unknown cases. If the increasing regions occur within each class, this routine will fail to work.

In summary, we claim that **current methods in distance metric learning can improve its discriminant ability and reduce the Lipschitz constant of the learned problem at the same time by applying the right regularization factor.** In this paper, we design a framework to demonstrate this. The contents include the following aspects:

1  We give an unified framework of proxy-based distance metric learning. In our framework, we summarize the traditional distance metric learning algorithms and classification-based distance metric learning algorithms together. Therefore, we present the mathematical framework that proves the connections between these methods and it gives us a theoretical base to support our hypothesis on the effects of the Lipschitz constant on distance metric learning.

2  We reveal that potential energy-based methods are not very effective in pushing centers away from each other since it only consider local information of the data. To alleviate this problem, we adopt the log-exp mean functions and power mean function to design a loss term that pulls away centers of each classes. Because we prove that potential energy methods are a special realization of our algorithm, ours also has the power to separate centers of different classes.

3  To further solve the Lipschitz constant problem on distance metric learning we design a deep neural network structure that allow us to minimize the Lipschitz constant of the deep neural network directly. This structure contains two parts: the first part extracts features using traditional backbone networks, such as Resnet, VGG, and Inceptions; the second

part learns the non-linear metric by an invertible deep neural layer used in the Normalizing Flows, whose gradients with respect to the input are easy to compute.

4 We conduct extensive experiments on challenging data sets such as CUB-200-2011, Cars196, Standord Online Products, and In-Shop Clothes Retrieval to illustrate the effectiveness of our algorithm.

**Notation.** We denote as $\mathcal{X}^o = \{(\mathbf{x}_i, y_i)\}_{i=1}^{N_1}$ the $C$-class dataset where $\mathbf{x}_i \in \mathbb{R}^{d_1}$ is the $i$-th sample and $y_i \in \{1, \cdots, C\}$ is the label of $\mathbf{x}_i$. $\mathbf{z}_i = f_\theta(\mathbf{x}_i) : \mathbb{R}^{d_1} \to \mathbb{R}^{d_2}$ is a deep neural networks parameterized by $\theta$. The similarity between $\mathbf{x}_i$ and $\mathbf{x}_j$ is denoted as $\mathbf{A}_{ij} = cos(f_\theta(\mathbf{x}_i), f_\theta(\mathbf{x}_j))$. The set of proxies is denoted by $\mathcal{X}^p = \{(\mathbf{w}_k, y_k)\}_{k=1}^{N_2}$ where $\mathbf{w}_k \in \mathbb{R}^{d_2}$ and $y_k \in \{1, \cdots, C\}$ is the corresponding label of $\mathbf{w}_k$. The similarity between $\mathbf{x}_i$ and $\mathbf{w}_j$ is denoted by $\mathbf{B}_{ij} = cos(f_\theta(\mathbf{x}_i), \mathbf{w}_j)$. Because proxy-based DML does not calculate the similarity between samples within $\mathcal{X}$ or $\mathcal{X}^p$, the similar relationship between samples $\mathcal{X}^o + \mathcal{X}^p$ can be depicted by a bipartite graph. For $\mathbf{x}_i \in \mathcal{X}^o$, its similar samples are only in $\mathcal{X}^p$ and denoted as $\mathcal{S}_i^1$. For $\mathbf{w}_i \in \mathcal{X}^p$, its similar samples are only in $\mathcal{X}^o$ and denoted by $\mathcal{S}_i^2$. Likewise, dissimilar sample sets of $\mathbf{x}_i \in \mathcal{X}^o$ and $\mathbf{w}_i \in \mathcal{X}^p$ are denoted by $\mathcal{D}_i^1$ and $\mathcal{D}_i^2$, respectively.

## 2 MOTIVATION

Let us consider a deep neural network $\mathbf{z} = f_\theta(\mathbf{x}) : \mathbb{R}^{d_1} :\to \mathcal{R}^d$ where $\theta$ is the learnable parameter, $\mathbf{x}$ is the image vector, and $\mathbf{z}$ is the feature vector of the image vector $\mathbf{x}$. Normally, in DML the norm of $\mathbf{z}$ is equal to 1 because the $L_2$-normalization is often used on the last layer of $f_\theta(\mathbf{x})$. Therefore, for a training dataset $\mathcal{X}_{tr} = \{(\mathbf{x}_i, y_i)\}_{i=1}^N$ and a testing data $\mathcal{X}_{te} = \{\mathbf{x}_i\}_{i=1}^T$, the samples of them are all projected to the surface of a $d$-dimensional sphere centered at the origin of the feature space.

Let us denote the surface of the considered $d$-dimensional sphere by $\mathcal{S}$. Because in classification-based tasks, the features vectors of different classes are desired to be separated from each other, the features from different classes are located in different clusters on $\mathcal{S}$. Without loss of generality, we suppose that each class of samples belong only to one cluster, and the cluster region located by the $k$-th class is denoted by $\mathcal{S}_k$. Therefore, the set $\mathcal{S}$ is divided into $C + 1$ parts. Besides the $C$ regions $\{\mathcal{S}_k\}_{k=1}^C$, there is one region without any samples located in, which is called as *blank region*. In the open set problem, the blank region is specified to the unknown classes. In distance metric learning, the blank region of the training features may be located within the region of the testing features. Because there is no overlapping between classes of the training and the testing sets, therefore, $\mathcal{S} = \mathcal{B} + \cup_{k=1}^n \mathcal{S}_k$.

Then, we would like to demonstrate why shrinking the samples will increase the Lipschitz constant of the learned projection. Before doing this, we introduce the definition of the Lipschitz constant.

**Definition.** Let $(\mathcal{X}, d^{\mathcal{X}})$ and $(\mathcal{Y}, d^{\mathcal{Y}})$ be two metric spaces, the Lipschitz constant of a function $f$ is defined as:

$$Lip(f) = \max_{\mathbf{x}_1, \mathbf{x}_2 \in \mathcal{X}: i \neq j} \frac{d^{\mathcal{Y}}(f(\mathbf{x}_1), f(\mathbf{x}_2))}{d^{\mathcal{X}}(\mathbf{x}_1, \mathbf{x}_2)} \tag{2}$$

Let us consider two projection $f_1(\mathbf{x})$ and $f_2(\mathbf{x})$, their projecting regions on $\mathcal{S}$ are $\{\mathcal{S}_k^1\}_{k=1}^C + \mathcal{B}^1$ and $\{\mathcal{S}_k^2\}_{k=1}^C + \mathcal{B}^2$, respectively. Thus, if the area of $\{\mathcal{S}_k^1\}_{k=1}^C$ is larger than the area of $\{\mathcal{S}_k^2\}_{k=1}^C$, the area of $\mathcal{B}^1$ is smaller than that of $\mathcal{B}^2$. Therefore, we can find two samples $\mathbf{x}_a$ $\mathbf{x}_b$ whose projections are in $\mathcal{B}^1$ and $\mathcal{B}^2$. Thus, the following constraint holds

$$\frac{d_Y(f_1(\mathbf{x}_a), f_1(\mathbf{x}_b))}{d_X(\mathbf{x}_a, \mathbf{x}_b)} < \frac{d_Y(f_2(\mathbf{x}_a), f_2(\mathbf{x}_b))}{d_X(\mathbf{x}_a, \mathbf{x}_b)} \tag{3}$$

The Eq. (3) indicates that the Lipschitz constant of the $f_1$ is smaller than that of $f_2$.

However, for distance metric learning tasks, the dimension of $\mathcal{S}$ is very small when compared to the number of training samples. For example, the CUB200-2021 dataset has 11,788 samples with 200 classes. Each class has less than 60 images on average. Thus, $\mathcal{S}_k$, the region of the $k$-th class may be not a connected to the neighborhood of another class, otherwise we consider that both belong to the same one. As seen from Figure 1, between two similar samples there could be a space belonging

to the blank region. Thus, when we shrink the distance between samples of each class, the regions involving the increasing Lipschitz constant may probably be within the blank areas between two similar samples. When this happens, the generalization ability of $f_1$ would be harmed.

For a good training on the deep neural network, we want to learn the distance metric learning without increasing the Lipschitz constant within the cluster of each class.

## 3 ANALYSIS ON THE LIPSCHITZ CONSTANT OF NEURAL NETWORK

In this section, we introduce how to control the Lipschitz constant of our deep network. Before doing this, we introduce an useful lemma about the Lipschitz constant of a deep layer-based projection.

**Lemma 1.**[Weaver (2018)] *Given a $T$-layer deep projection $f_\theta(\boldsymbol{x}^0) = f_T(f_{T-1} \ldots f_1(\boldsymbol{x}^0))$ parameterized with $\theta$, the $i$-th layer is $\boldsymbol{x}^{i+1} = f_i(\boldsymbol{x}^i)$. Let $L_{f_i}$ and $L_{f_\theta}$ denote the Lipschitz constant of $f_i(\boldsymbol{x}^i)$ and $f_\theta(\boldsymbol{x}^0)$, there is an equation of $L_{f_\theta}$ constrained by*

$$L_{f_\theta} = \prod_{i=1}^{T} L_{f_i} \tag{4}$$

According to Eq.(4), the Lipschitz constant of $f_\theta(\mathbf{x})$ can be constrained calculating the individual contribution of each layer $\{L_{f_i} | i = 1, \cdots, T\}$. Let us introduce another lemma which gives a tighter upper bound to the Lipschitz constant of the projection $L_{f_i}$.

**Lemma 2.** *Given a deep neural network, the Lipschitz constant of its $i$-th layer presents the following relationship.*

$$L_{f_i} = \max_{\mathbf{x}_1, \mathbf{x}_2 \in \mathcal{X}} \frac{\|f_i(\mathbf{x}_1) - f_i(\mathbf{x}_2)\|^2}{\|\mathbf{x}_1 - \mathbf{x}_2\|^2} = \max_{\mathbf{x}' \in \mathcal{X}} \|(\frac{\partial f_i}{\partial \mathbf{x}})|_{\mathbf{x}=\mathbf{x}'}\|_F^2 = \max_{\mathbf{x}' \in \mathcal{X}} \sum_{i=1}^{d} (\lambda_i^{\mathbf{x}'})^2 \tag{5}$$

where $\lambda_i^{\mathbf{x}'}$ is the $i$-th singular value of the matrix $\frac{\partial f_i}{\partial \mathbf{x}}|_{\mathbf{x}=\mathbf{x}'}$.

**Proof**. Suppose $f_i$ is a continuous function, according to the Taylor equation there is a $\mathbf{x}'$ such that $f_i(\mathbf{x}_2) = f_i(\mathbf{x}_1) + \mathbf{A}_{\mathbf{x}'}(\mathbf{x}_2 - \mathbf{x}_1)$, where $\mathbf{A}_{\mathbf{x}'} = \frac{\partial f_i(\mathbf{x})}{\partial \mathbf{x}|}_{\mathbf{x}=\mathbf{x}'}$. Thus, $\max_{\mathbf{x}_1, \mathbf{x}_2} \frac{\|f_i(\mathbf{x}_1) - f_i(\mathbf{x}_2)\|^2}{\|\mathbf{x}_1 - \mathbf{x}_2\|^2} = \max_{\mathbf{x}', \mathbf{x}_1, \mathbf{x}_2} \frac{(\mathbf{x}_2 - \mathbf{x}_1)^T \mathbf{A}_{\mathbf{x}}^T \mathbf{A}_{\mathbf{x}} (\mathbf{x}_2 - \mathbf{x}_1)}{\|\mathbf{x}_1 - \mathbf{x}_2\|^2} = \max_{\mathbf{x}'} \|\mathbf{A}_{\mathbf{x}}\|_F^2$. Denoting the singular values of $\mathbf{A}_{\mathbf{x}'}$ by $\{\lambda_i^{\mathbf{x}'}\}_{i=1}^d$, there is $\|\mathbf{A}_{\mathbf{x}}\|_F^2 = Tr(\mathbf{A}_{\mathbf{x}}^T \mathbf{A}_{\mathbf{x}}) = \sum_{i=1}^{d} (\lambda_i^{\mathbf{x}'})^2$. Thus, $L_{f_i} = \max_{\mathbf{x}' \in \mathcal{X}} \sum_{i=1}^{d} (\lambda_i^{\mathbf{x}'})^2$. □

**Remark 1.** *The above **Lemma** estimates the Lipschitz constant of a projection by calculating partial gradient of $f_i(\boldsymbol{x})$ with respect to $\boldsymbol{x}$. Thus, if we reduce the F-norm of the partial gradient matrix at all the samples in the training dataset, we can let the learned projection have a smaller Lipschitz constant. If $f_i$ represents a linear projection: $\boldsymbol{y} = \boldsymbol{L}^T \boldsymbol{x}$. Because $\frac{\partial f_i}{\partial \boldsymbol{x}} = \boldsymbol{L}^T$, the Lipschitz constant corresponds to $L_{f_i} = \|\boldsymbol{L}\|_F^2$ which is a widely-used regularization term to improve the generalization ability on deep learning models.*

Distance metric learning learns a representation where samples in the same class present a small distance, and samples from different classes a large distance. Thus, for the deep neural networks $f_\theta$ trained on this metric, the lipschitz constant of $f_\theta$ will naturally increase in the black region and decrease in clusters. However, if we use the term in Eq.(5) to minimize the Lipschitz constant of $f_\theta$, then we would reduce the Lipschitz constant of the overall problem. Such a result contradicts the goal of distance metric learning. Because it harms the discriminant ability of the model, but in return it improves generalization.

Thus, instead of using $\sum_{i=1}^{d} \lambda_i^x (dx_i)$, we introduce the following term to regularize the Lipschitz constant.

$$\mathbb{R}_{x'} = log(\prod_{i=1}^{d} (\lambda_i^{x'})^2) = 2 \sum_{i=1}^{d} log(\lambda_i^{x'}) \tag{6}$$

**Geometric meaning .** Suppose $\mathbf{A}_{\mathbf{x}}$ is the Jacobian matrix of $\mathbf{z} = f_i(\mathbf{x})$ at $\mathbf{x}'$. $O(\mathbf{x}, r) = \{\mathbf{x}||\mathbf{x}' - \mathbf{x}| < r\}$ is a neighborhood of $\mathbf{x}'$, so the volume of $O(\mathbf{x}, r) = \prod_{i=1}^{d} (dx_i)$ if $r \to 0$. Suppose the singular

values of $\mathbf{A_x}$ are $\lambda_1^x > \lambda_2^x > \cdots > \lambda_d^x$. The volume of $f_i(O(\mathbf{x}, r))$ is $\prod_{i=1}^d \lambda_i^x (dx_i)$, thus, $\prod_{i=1}^d \lambda_i^x$ is the volume changed after projection.

Therefore, if $\mathbb{R}_{x'}$ is reduced, the value $\|\mathbf{A}_{x'}\|_F^2 = \sum_{i=1}^d (\lambda_i^{x'})^2$ is reduced. But the difference from $\|\mathbf{A}_{x'}\|_F^2$ is that minimizing $\mathbb{R}_{x'}$ will let small $(\lambda_i^{x'})^2$ become smaller at the procedure of minimizing $\|\mathbf{A}_{x'}\|_F^2$.

Therefore, we can propose a regularization term to minimize the Lipschitz constant according to the metric learning requirement.

$$R(\mathbf{x}_i) = \frac{1}{d} \sum_{i=1}^d log\left(\lambda_i^{x'} + 1\right) \tag{7}$$

## 4 DEEP METRIC LAYER

From the implementation point of view, the formulation involved in the calculus of $det(\frac{\partial f_\theta(\mathbf{x})}{\partial \mathbf{x}})$ is intractable for traditional neural networks. To alleviate this problem, we design an non-linear projection layer for whom the determinant of its Jacobian matrix is easy to solve. Here, we adopt the deep neural network used in Normalizing Flows Rezende & Mohamed (2015) where it is efficient to solve $det(\frac{\partial f_\theta(\mathbf{x})}{\partial \mathbf{x}})$.

Let $h(\cdot; \theta) : \mathbb{R} \to \mathbb{R}$ be a bijection parameterized by $\theta$. Then, the desired projection is $\mathbf{g} : \mathbb{R}^D \to \mathbb{R}^D$, which projects each sample $\mathbf{x} \in \mathbb{R}^D$ as $\mathbf{y} = g(\mathbf{x})$. Let the $t$-th entries of $\mathbf{x}$ and $\mathbf{y}$ be $x_t$ and $y_t$, the projection is defined as

$$y_t = h(x_t; \Theta_t(\mathbf{x}_{1:t-1})) \tag{8}$$

where $\mathbf{x}_{1:t} = (x_1, \cdots, x_t)$. For $t = 2, \cdots, D$ we choose arbitrary functions $\Theta_t(\cdot)$ mapping $\mathbb{R}^{t-1}$ to the set of all parameters, and $\Theta_1$ is a constant.

The jacobian matrix of the Eq.(9) is triangular. Each output $y_t$ only depends on $\mathbf{x}_{1:t}$ and so the determinant is just the product of its diagonal entries,

$$det(Dg) = \Pi_{t=1}^D det\left(\frac{\partial y_t}{\partial x_t}\right) \tag{9}$$

The structure of Normalizing Flows is an invertible projection which can not be used to reduce the dimension of the input samples. So the input of the Normalizing Flows can not be images, thus we need employ a backbone based on convolutional neural networks to extract the features of images and at last those features are fed into the Normalizing Flow layers. Therefore, the regularization presented in the previous section can not be used to constrain the Lipschitz constant of the convolutional neural networks. Let us consider the Eq.(5), we find that if we only constrain the Lipschitz constant of the structure of Normalizing Flows, two results may occur. (1) the Lipshcitz constant of whole structure of our deep neural network will be reduced. (2) the Lipschitz constant of the convolutional neural works will be larger. To avoid the second situation, we use a dueling Normalizing Flows architecture connecting to the backbone. For the first one, the regularization is minimized, while for the second one, its regularization term is maximized. During the inference, only the minimizing deep metric layer are used. By doing this, we can let the Lipschitz constant of the backbone be stable while reducing Lipschitz constant of the combined network. The Structure of the proposed method is depicted in the Figure 2. According to the property of the invertible structure of normalizing flow, the Jacobi matrix $\frac{\partial y_t}{\partial x_t}$ is an upper triangle matrix. Thus, there is an equation that

$$R(\mathbf{x}_i) = \frac{1}{d} \sum_{i=1}^d log\left(\lambda_i^{x'} + 1\right) == \frac{1}{d} log\left(\frac{\partial y_t}{\partial x_t} + \mathbf{I}\right) \tag{10}$$

where $\mathbf{I}$ is an identity matrix.

Because the $R(\mathbf{x}_i)$ is a function of $\mathbf{x}_i$. If we have the sample set $\mathcal{Z} = \{z_1, \cdots, z_n\}$, then we can use $R(\mathcal{Z}) = \frac{1}{n} \sum_{i=1}^n R(\mathbf{z}_i)$.

The Eq.(11) lets the regularization of the Lipscthiz constant be small for each training sample. However it does not reduce the Lipscthiz constant between dissimilar samples. To solve this problem, we use sample augmentation to reduce Lipschitz constant in those regions.

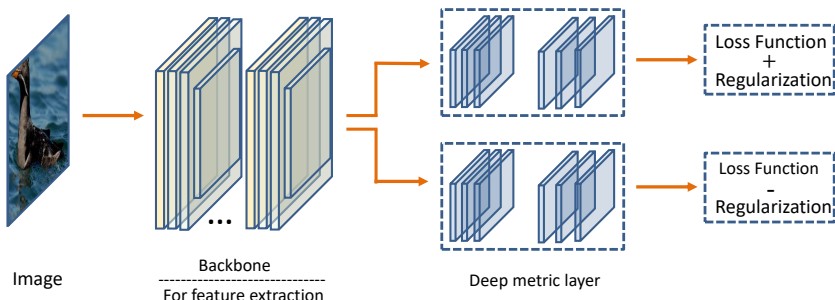

Figure 2: Structure of the dueling deep metric neural network used to reduce the overall Lipschitz constant of the network.

**Sample augmentation.** Let $\mathbf{z}_i = f_\theta(\mathbf{x}_i)$ be the output feature embedding of the backbone. Then, for a point $\{\mathbf{z}_i\}$, we can be compute the direction $\mathbf{e}_{ij} = \mathbf{z}_i - \mathbf{z}_j$. Then, a new sample can be generated sampling from $\mathbf{z}_{ij} = \mathbf{z}_i + \varepsilon\mathbf{e}_{ij}$. By selecting an appropriate value $\varepsilon$, we can find that the new sample $\mathbf{z}_{ij}$ belongs to the blank region between $\mathbf{z}_i$ and $\mathbf{z}_j$. In this way, if we minimize the Lipschitz constant on the generated samples, we can reduce the Lipscthiz constant of points within the cluster of each class. In order to efficiently select the number of samples, we only select the samples $\mathbf{x}_j$ in the neighborhood of $\mathbf{x}_i$. In this way, we produce extra training samples for the batch. Because those augmented samples are located between two similar samples, if the distance between the samples is reduced, the distance between the augmented sample and each of the two similar ones also does. Therefore, there is no need to perform the distance metric learning on the augmented samples.

Thus, the objective function of the proposed distance metric learning is presented as follows.

$$\min_{(\theta_1, \theta_2, \theta_3)} \sum_{i=1}^{N} loss((\mathbf{x}_i, \mathcal{S}_i, \mathcal{D}_i), \theta) + \lambda_1 R_{\theta_2}(\mathcal{Z}_1) - \lambda_2 R(\mathcal{Z}_2) \tag{11}$$

where $\theta_1$ is the learnable parameter of the backbone, $\theta_2$ is the learnable parameter of the parameter of the first deep metric layer, $\theta_3$ is the second deep metric layer. $\lambda_1 > 0$ and $\lambda_2 > 0$ are the coefficient factors of the regularization terms. $\mathcal{Z}_1$ is the training samples and their augmented samples for the first invertible neural network, and $\mathcal{Z}_2$ is the training samples and the augmented samples for the second invertible neural network.

After the training of the model, the outputs of the first invertible networks are considered as the features of images.

## 5 EXPERIMENTS

We evaluate the effectiveness of the proposed method on four datasets for fine-grained image retrieval. In the conducted experiments we compare the performance of the proposed regularization factor when applied on current state-of-the-art models in DML.

### 5.1 SETTINGS

**Fine-grained image retrieval.** We benchmark our model in four datasets on fine-grained image retrieval: Cars196, CUB-200-2011, Standord Online Products (SOP) and In-Shop Clothes Retrieval. CARS-196 contains 16,183 images from 196 classes of cars. The first 98 classes are used for training and the last 98 classes are used for testing. CUB-200-2011 contains 200 different class of birds. We use the first 100 class with 5,864 images for training and the last 100 class with 5,924 images for testing. SOP is the largest dataset and consists of 120,053 images belonging to 22,634 classes of online products. The training set contains 11,318 classes and includes 59,551 images, the rest 11,316 classes with 60,499 images are used for testing. Lastly, the In-shop dataset consists of a total of 54,642 images which are divided in 25,882 images from 3997 classes for training and 28,760 images for testing.

Table 1: Comparison of the Recall@k (in percentile) on the CUB-200-2011 and the Cars-196 fine-grained image datasets. The backbone network is denoted by: G for GoogleNet, R50 for ResNet50 and BN for Inception with Batch Normalization. The superscript indicates the size of the final embedding layers used in the backbone network. *Source:* Kim et al. (2020).

| | | CUB-200-2011 | | | | Cars-196 | | | |
|---|---|---|---|---|---|---|---|---|---|
| *Recall* | | *k=1* | *k=2* | *k=4* | *k=8* | *k=1* | *k=2* | *k=4* | *k=8* |
| Clustering[64] | BN | 48.2 | 61.4 | 71.8 | 81.9 | 58.1 | 70.6 | 80.3 | 87.8 |
| Proxy-NCA[64] | BN | 49.2 | 61.9 | 67.9 | 72.4 | 73.2 | 82.4 | 86.4 | 87.8 |
| Smart Mining[64] | G | 49.8 | 62.3 | 74.1 | 83.3 | 64.7 | 76.2 | 84.2 | 90.2 |
| MS[64] | BN | 57.4 | 69.8 | 80.0 | 87.8 | 77.3 | 85.3 | 90.5 | 90.2 |
| SoftTriple[64] | BN | 60.1 | 71.9 | 81.2 | 88.5 | 78.6 | 86.6 | 91.8 | 95.4 |
| Proxy-Anchor[64] | BN | 61.7 | 73.0 | 81.8 | 88.8 | 78.8 | 87.0 | 92.2 | 95.5 |
| Margin[128] | R50 | 63.6 | 74.4 | 83.1 | 90.0 | 79.6 | 86.5 | 91.9 | 95.1 |
| HDC[384] | G | 53.6 | 65.7 | 77.0 | 85.6 | 73.7 | 83.2 | 89.5 | 93.8 |
| A-BIER[512] | G | 57.5 | 68.7 | 78.3 | 86.2 | 82.0 | 89.0 | 93.2 | 96.1 |
| ABE[512] | G | 60.6 | 71.5 | 79.8 | 87.4 | 85.2 | 90.5 | 94.0 | 96.1 |
| HTL[512] | BN | 57.1 | 68.8 | 78.7 | 86.5 | 81.4 | 88.0 | 92.7 | 95.7 |
| RLL-H[512] | BN | 57.4 | 69.7 | 79.2 | 86.9 | 74.0 | 83.6 | 90.1 | 94.1 |
| MS[512] | BN | 65.7 | 77.0 | 86.3 | 91.2 | 84.1 | 90.4 | 94.0 | 96.5 |
| SoftTriple[512] | BN | 65.4 | 76.4 | 84.5 | 90.4 | 84.5 | 90.7 | 94.5 | 96.9 |
| Proxy-Anchor[512] | BN | 68.4 | 79.2 | 86.8 | 91.6 | 86.1 | 91.7 | 95.0 | 97.3 |
| Ours[512] | BN | 69.1 | 80.1 | 87.2 | 92.2 | 87.2 | 92.2 | 95.3 | 97.4 |

## 5.2 IMPLEMENTATION DETAILS

We use Inception as the backbone for our model. Similar to previous works in the literature, we use a pre-trained model on ImageNet for classification and we select a final 512-D embedding layer that would correspond to the dimension of the hidden layers on the deep metric model. For the backbone we freeze the batch normalization layers during the training and we add an activation layer for the connection to the deep metric layers. The deep metric layer we use are invertible normalizing flows layers, as the determinat of the Jacobian matrix required in the regularization factor is efficient to compute. Particularly, we rely on Real-NVP Dinh et al. (2016), a model that implements normalizing flows using affine coupling layers that combine a scaling term with a shit term in the transformation. Despite its simplicity Real-NVP has shown to be effective estimating complex density distributions without requiring a high number of layers. In our experiment we rely on 12 layers for each of the dueling deep metric modules to capture the radial distribution proceed by the cosine similarity in DML. Finally, we use $L_2$-normalization on the final output of the normalizing flows.

Regarding the loss function, we rely on the state-of-the-art Proxy Anchor loss Kim et al. (2020) for our experimentation. Proxy Anchor proposes a proxy-based anchor method that associates the entire data in the batch with proxies for each class. This method has shown advantages w.r.t. previous proxy-based method that do not exploit data-to-data relations and it has also shown to be more efficient than pair-based methods. In particular, we define the same number of proxies as classes in the dataset. The loss function is completed with the determinant-based regularization factor presented Eq. (11).

**Training:** To train our model we rely on the weight decay AdamW optimizer Loshchilov & Hutter (2017). The initial learning rate is $10^{-4}$ for the backbone network and $10^{-3}$ for the metric learning layers and the proxies. We use a linear decay in both cases and we train the model for 30 epochs. Similarly to Kim et al. (2020) we initialize the proxies using a normal distribution and a bigger learning rate is used on them for a faster convergence.

We maintain the hyperparameters $\delta$ for the margin and $\alpha$ scaling factor of Proxy Anchor to 32 and 0.1, respectively, in all experiments. We set the coefficients $\lambda_1$ and $\lambda_2$ to 0.05 for the dueling deep metric layers.

Table 2: Comparison of the Recall@k (in percentile) on the Standord Online Products (SOP).

| Recall | k=1 | k=10 | k=100 | k=1000 |
|---|---|---|---|---|
| | **SOP** | | | |
| Clustering[64] | 67.0 | 83.7 | 93.2 | - |
| Proxy-NCA[64] | 73.7 | - | - | - |
| MS[64] | 74.1 | 87.8 | 94.7 | 98.2 |
| SoftTriple[64] | 76.3 | 89.1 | 95.3 | - |
| Proxy-Anchor[64] | 76.5 | 89.0 | 95.1 | 98.2 |
| Margin[128] | 72.7 | 86.2 | 93.8 | 98.0 |
| HDC[384] | 69.5 | 84.4 | 92.8 | 97.7 |
| A-BIER[512] | 74.2 | 86.9 | 94.0 | 97.8 |
| ABE[512] | 76.3 | 88.4 | 94.8 | 98.2 |
| HTL[512] | 74.8 | 88.3 | 94.8 | 98.4 |
| RLL-H[512] | 76.1 | 89.1 | 95.4 | - |
| MS[512] | 78.2 | 90.5 | 96.0 | 98.7 |
| SoftTriple[512] | 78.3 | 90.3 | 95.9 | - |
| Proxy-Anchor[512] | 79.1 | 90.8 | 96.2 | 98.7 |
| Ours[512] | 79.0 | 90.7 | 96.1 | 98.7 |

Table 3: Comparison of the Recall@k (in percentile) on the In-Shop Clothes Retrieval dataset.

| Recall | k=1 | k=10 | k=100 | k=1000 |
|---|---|---|---|---|
| | **In-shop** | | | |
| HDC[384] | 62.1 | 84.9 | 89.0 | 92.3 |
| HTL[128] | 80.9 | 94.3 | 95.8 | 97.4 |
| MS[128] | 88.0 | 97.2 | 98.1 | 98.7 |
| Proxy-Anchor[128] | 90.8 | 97.9 | 98.5 | 99.0 |
| FashionNet[4096] | 53.0 | 73.0 | 76.0 | 79.0 |
| A-BIER[512] | 83.1 | 95.1 | 96.9 | 97.8 |
| ABE[512] | 87.3 | 96.7 | 97.9 | 98.5 |
| MS[512] | 89.7 | 97.9 | 98.5 | 99.1 |
| Proxy-Anchor[512] | 91.5 | 98.1 | 98.8 | 99.1 |
| Ours[512] | 91.5 | 98.2 | 98.7 | 99.1 |

## 5.3 RESULTS

To evaluate the benefits of the proposed regularization factor we follow Kim et al. (2020) and perform an experimentation on the CUB200-2011, Cars-196, Standord Online Products and In-Shop Clothes Retrieval datasets for an image size of 224×224 pixels.

The results for the CUB-200-2011 and the Cars-196 datasets are summarize in Table 1. Our method achieves competitive results on both dataset, improving the recall@k metric of the baseline implementation of Proxy Anchor. Significant results are obtained on both cases, where we beat by a margin of 0.7% the recall@1 in the CUB-200-2011 dataset and by a margin of 1.1 in the Cars-196. We recreate the Table 1 for Kim et al. (2020) to put in perspective the obtained results. In the SOP dataset, the model shows slightly worse performance that the baseline, the results are summarized in Table 2. We attribute these results to the fact that the SOP dataset is the largest of the four. Despite this fact we expect that with further tunning of the parameters this result can be also improved. At last, the results on the In-shop dataset are also modest in comparison to the baseline scenario (see Table 3), and we cannot attribute a significant benefit on applying the regularization factor in that case.

## 5.4 ABLATION STUDIES

**Deep metric layers.** In the ablation studies we test two types of Normalizing Flows for the distance metric layers: Real-NVP Dinh et al. (2016) and NICE (Non-linear Independent Component Estimation) Dinh et al. (2014) considered the predecessor of Real-NVP. We observe a slight benefit in using Real-NVP and therefore this is the metric that we report in the experimentation.

**Coefficients of regularization.** Regarding the regularization coefficient, we perform a test comprising the following values of $\lambda \in \{0.01, 0.05, 0.1, 0.2, 0.5\}$. Particularly, we observe better results when the coefficients $\lambda_1$ and $\lambda_2$ of the dueling deep metric layers take the same value. The reported results correspond to a value of $\lambda_1 = \lambda_2 = 0.05$ that achieved overall best performance.

**Data augmentation.** Finally we also experiment with different values for the $\epsilon$-coefficient in the generation of new samples for data augmentation. The $\epsilon$-coefficient controls how far from the current sample is the augmented one be created. Because the last layer applies an $L_2$-normalization, the output samples are restricted to be in the unitary hyper-sphere and the data augmentation occurs in the tangent hyperplane to it. We test several coefficients $\epsilon \in \{0.01, 0.05, 0.1, 0, 2, 0.4, 0.6, 0.8\}$ ranging from the close proximity of the original sample to the neighbour sample. In order to further increase the generalization we should carefully select the $\epsilon$, in our experimentation $\epsilon = 0.4$ corresponds to the value that achieved best performance.

We keep the ablation studies concise because the aim in the experimentation is to determine whether the regularization benefits the proxy-anchor algorithm used as a baseline. Therefore, in the shake of presenting a fair comparison, we maintain the same 512-D embedding dimension as the original work does, and also we report the same values for the margin $\delta = 0.1$ and scaling factor $\alpha = 32$ of Proxy Anchor.

## 6 CONCLUSIONS

This paper presents a novel learning paradigm for Distance Metric Learning (DML). Differently from the other DML methods that only focus on designing different loss functions, our work focuses on regularizing the Lipschitz constant as a way to improve the generalization capabilities of DML models. We adopt invertible layers from Normalizing Flows to construct a deep metric model in which the computation the Jacobian matrix is efficient. At last, we minimize the determinant of Jacobian matrix to reduce the Lipschitz constant of the deep neural network. Conducted experiments on fine-grained Cars196, CUB-200-2011, Standord Online Products (SOP) and In-Shop Clothes Retrieval datasets show that the proposed architecture benefits the baselines proxy-based architecture on achieving better generalization.

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
