# OpenReview forum: "Determinant regularization for Deep Metric Learning"
_ICLR.cc/2023/Conference — Submitted to ICLR 2023_

### Official Review · Reviewer_VbA6 · 2022-10-14

**Confidence:** 2
**Correctness:** 2
**Technical Novelty And Significance:** 3
**Empirical Novelty And Significance:** 2
**Recommendation:** 3

**Clarity, Quality, Novelty And Reproducibility:**

As I discussed above, the clarity of this paper is good. The idea of using normalizing flow to reduce the Lipschitz constant is also new to me. However, it seems that this paper has some incorrect claims, so I think its quality may not be that high.

**Strength And Weaknesses:**

**Strengths:**

(1). This paper is clearly written and easy to follow.

(2). The high-level motivation is good. I agree with the authors that excessively reducing intra-class distance may incur over-fitting.

(3). Theoretical analysis is provided.


**Weaknesses:**

This paper has two critical flaws.

(1). The authors claim that the L2 normalization is only available for linear layers, so they cannot be used in deep metric learning. I don't think so. Actually, there is a weight decay technique in deep learning, which also effectively reduces the Lipschitz constant of the neural network. In this case, the motivation for proposing a new regularization technique in this paper is not solid.

(2). For pair-based loss functions such as contrastive similarity loss, I agree with the authors that they may aim to reduce the intra-class distance as much as possible. However, for many relative similarity loss functions, we only encourage that it is okay when inter-class distances are larger than intra-class distances. We would not excessively reduce the intra-class distances as much as possible.

(3). Furthermore, I found the compared methods in experiments are before 2020, which means they cannot be regarded as sota.

**Summary Of The Paper:**

This paper considers the over-fitting issue caused by reducing intra-class distances in metric learning. Specifically, when the intra-class distances are excessively reduced, the Lipschitz constant of network embedding would be significantly enlarged. Then the decision boundary comes unsmooth and is not good for generalization performance. To address this issue, the authors propose a normalizing flow as a new network layer to regularize the learning. Experiments on popular datasets validate the effectiveness of the proposed method.

**Summary Of The Review:**

Overall, this paper is well-written and the idea is novel, but I do not agree with some critical claims in this paper. Since these claims are rather critical to the motivation, I would like to vote for a "reject", and I think the authors should further clarify them in the new version.

---

### Official Review · Reviewer_obNj · 2022-10-21

**Confidence:** 4
**Correctness:** 3
**Technical Novelty And Significance:** 2
**Empirical Novelty And Significance:** 2
**Recommendation:** 1

**Clarity, Quality, Novelty And Reproducibility:**

Clarity
- A lot of sentences are unclear. The paper requires polishing. I can refer to the second paragrpah of section 2 for instance or the first sentence of section 4.
- Some mathematical terms are inconsistent or not introduced. d2 is used as output dimensionality in the notation and never used after. Similarly, proxies are mentipnned in the notation part but never used later. The Blank region is annotated by B by never clearly stated, we have to read a second time to understand.
- Overall, the mathematical notation could be improved.
Quality
- The lack of ablation study tables showing the effect of different design choices is deteriotating the quality of the paper.
Reproducibility
The overall idea of the paper comes through and seems reproducible. The theory behind the manipulation and approximation of the Lipschitz constant seems grounded.

**Details Of Ethics Concerns:**

There is no Ethic concerns

**Strength And Weaknesses:**

Strengths:
- The strength of this paper resides in the strong theoritical analysis of the effect the Lipschitz constant has on the embedding function modelled by the deep neural network.
- The authors also show some improvement on the CUB-200-2011 and Cars-196 Dataset
- The utilization of Normalizing flow network seems to be appropriate for approximating the Jacobian.

Weaknesses:
-  The main weakness of this paper is that it seemed to have been unfinished. They are a lot of typos and mistakes that degrades the overall quality of the paper. Most importantly the ablation study tables are missing.
- Incremental contribution. The main contribution is not significant enough in terms of motivation and performance. While the regularization term shows improvement on 2 datasets, it does not on 2 others that are much larger and more complicated. The performances seems to be mostly achieved by the baseline.
- The figure 2. is not informative, there is no disctinction between the 2 dueling branches. The captions are lacking information. What makes the top branch different from the bottom one?
- The main contribution is the presentation of  the regularization term where the Lipschitz constant is approximated and minimized for similar sample (as stated end of section 4 and with Eq. 11).
- In the conclusion the authors mention that other work focus on loss function design and not them. I would argue that design of regularization terms fall within the design of loss functions.

**Summary Of The Paper:**

The paper is about deep metric learning for fine-grained image retrieval. The task consists in learning a parameterized embedding function modelled as a neural network to project images into a deep space where images from the same categories are similar and images from different categories are not similar. The authors propose to train their model with a regularization term minimizing the Lipschitz constant around a set of data points from the same class. They approximate the Lipschitz constant using a Normalizing-Flow network.

**Summary Of The Review:**

I would not accept this paper because too many elements needs to be changed and the contribution is too incremental.

---

### Official Review · Reviewer_n2tf · 2022-10-24

**Confidence:** 3
**Correctness:** 1
**Technical Novelty And Significance:** 2
**Empirical Novelty And Significance:** 2
**Recommendation:** 3

**Clarity, Quality, Novelty And Reproducibility:**

I think this paper is good at clarity and reproducibility. The organization of this paper is reasonable and the expression is clear. Furthermore, the implementation details are very clear and those experimental results are supposed to be easy to be reproduced. The novelty and quality of this paper are ordinary, as the LC-based methods were studied to improve the robustness of neural networks before and the performance of this paper is not competitive.

**Strength And Weaknesses:**

Pros:
This paper focuses on a significant topic.
The organization of this paper is reasonable and the expression is clear, which make it easy to follow.

Cons.
The authors summarized four contributions of this paper, however, I did not find the first two.
I do NOT think the motivation of this paper holds, which is the crucial concern from me about this paper. This paper claims that shrinking the feature space will increase the Lipschitz constant (LC), and give some deductions in Eq (2) and Eq(3). It’s true you can find some x that satisfy Eq (3), but not for all x. Therefore, this can not prove the increasing or decreasing property of LC. Since the motivation does not hold, I do not see the meaning of the proposed method.
There are unclear statements, such as “the first/second deep metric learning layer”, “the learnable parameter of the parameter of”, “may be not a connected to the neighborhood of another class” which should be “may be connected to the neighborhood of another class” in my opinion.
The experimental part is very confusing. The proposed method compares with other methods based on the different backbone and embedding dimensions, which I don’t think can be compared together. Since your main strategy is to modify the learning objection, by adding a new regularization term, I advise you to combine multiple loss functions with your regularization term, which will be more convincing.

**Summary Of The Paper:**

This paper focuses on the deep metric learning field. The motivation of this paper is that Current pair-based and proxy-based methods harm the generalization ability as shrink the distance between positive pairs. As a result, it adopts the structure of normalizing flow as the deep metric layer and calculates the determinant of the Jacobian matrix as a regularization term that helps in reducing the Lipschitz constant.

**Summary Of The Review:**

The motivation of this paper doesn’t make sense to me, which is my main concern about it, hence I recommend rejection.

---

### Official Review · Reviewer_F9Bh · 2022-10-25

**Confidence:** 5
**Correctness:** 3
**Technical Novelty And Significance:** 3
**Empirical Novelty And Significance:** 2
**Recommendation:** 5

**Clarity, Quality, Novelty And Reproducibility:**

My main concern is the performance of the proposed method.  There are many SOTA works [1][2][3] or more under the same setting and backbones that the authors do not list for comparison, which is better than the results of R@1 the authors propose.

[1] Ebrahimpour et al. "Multi-Head Deep Metric Learning Using Global and Local Representations." CVPR 2022

[2] Zhang, Borui, et al. "Attributable Visual Similarity Learning." CVPR 2022

[3] Venkataramanan, Shashanka, et al. "It takes two to tango: Mixup for deep metric learning." ICLR 2022

**Strength And Weaknesses:**

Strength:
- Clear motivation and identification of the problem in existing methods.
- Proposed method is easy to understand and the description is clear.
- Theoretical proof is sufficient and complete.

Weaknesses:
- Experiments are not convincing:
1. An ablation experiment should be added to analyze the validity and necessity of each block.
2. The effectiveness of sample augmentation is necessary to discuss.
3. Visualization results should also be added.
4. The performances of the proposed methods are not good enough.
- Why Eq.(6) is better than Eq.(5)?


**Summary Of The Paper:**

This paper summarizes the traditional distance metric learning algorithms and classification-based distance metric learning algorithms together and solves the Lipschitz constant problem on distance metric learning. The authors design a deep neural network structure that allows us to minimize the Lipschitz constant of the deep neural network directly. They also minimize the determinant of the Jacobian matrix to reduce the Lipschitz constant of the deep neural network.

**Summary Of The Review:**

The idea of this paper show some novelty, but the overall writing quality is limited. Also, the experimental result is not convincing and competitive.

---

### Decision · Program_Chairs · 2023-01-20

**Decision:**

Reject

**Justification For Why Not Higher Score:**

All reviewers agree that this paper is a clear reject and the authors do not provide responses to reviewers' questions.

**Justification For Why Not Lower Score:**

N/A

**Metareview: Summary, Strengths And Weaknesses:**

All reviewers agree that this paper is a clear reject and the authors do not provide responses to reviewers' questions.